

# Propagating Information Content: An Example with Advection

David D. Turner[1], Maria P. Cadeddu[2], Julia Simonson[3,1], Timothy J. Wagner[4]

[1]NOAA / Global Systems Laboratory, Boulder, CO, USA
[2]Argonne National Laboratory, Argonne, IL, USA
[3]Cooperative Institute for Research in the Environmental Sciences, University of Colorado, Boulder, CO, USA
[4]Space Science and Engineering Center, University of Wisconsin – Madison, Madison, WI, USA

*Correspondence to*: David D. Turner (dave.turner@noaa.gov)

## Abstract:

The mathematical algorithm to derive geophysical information from remote sensing observations is called a retrieval. The mathematics of many retrieval problems is ill-posed, and thus a priori information is used to help constrain the derived geophysical variable to realistic values. One quantity of interest, therefore, is the information content of the observation. Perfect information content in the observation would be achieved if the retrieval is able to capture any perturbation in the desired geophysical variable with the proper magnitude.

Many new data products can be derived by combining geophysical variables retrieved from multiple different remote sensors. This paper explores, for the first time, how to derive the information content of these derived products. The approach uses traditional error propagation techniques to derive the uncertainty of the derived field twice, both when the observations are used in the retrieval and also when only the a priori information from each remote sensor is propagated. These two uncertainties are then used to provide an estimate of the information content of the derived geophysical variable.

This study demonstrates how to propagate the uncertainties from six different instruments to provide the information content for water vapor and temperature advection. A multi-month analysis demonstrates that, in a mean sense, the information content for temperature advection is nearly unity for all heights below 700 m while the information content for water vapor advection is somewhat more variable but still larger than 0.6 in the convective boundary layer.

## 1 Introduction

Observations are absolutely essential for science and understanding nature. They can serve both as the source of ideas (e.g., "This is an interesting observation; I wonder what it means?") and means to evaluate hypotheses (e.g., "My model suggests this is true; can I make an observation that confirms that the model is correct?"). In both of these cases, it is critical to understand the uncertainty in the observation in order to both correctly interpret the result.

Observations used in the natural sciences take advantage of many different physical principles. Some of these instruments
are considered 'in-situ'; in other words, the instrument makes its measurement of the desired geophysical variable at the





point of interest. Other instruments are remote sensors, where the instrument is displaced some distance from the location of interest. Remote sensors come in two general types: (a) active remote sensors, wherein the instrument transmits some signal such as electromagnetic energy or sound towards the measurement volume and analyzes the portion of the energy that is scattered from the volume towards the detector; and (b) passive remote sensors, which observe scattered or emitted signals

(typically electromagnetic radiation) from the measurement volume.

Seldom do we measure the actual geophysical variable that we desire; instead, virtually all instruments measure a signal that provides information that is related by some physical process to the variable we desire (Maahn et al. 2020). For example, a simple mercury-based thermometer provides a measure of temperature as the depth of the mercury in a vacated tube is directly proportional to the temperature due to the thermal expansion of the mercury in the reservoir; characterizing the

measurement uncertainty for instruments like this is reasonably straightforward.

Deriving geophysical variables from the observations made by remote sensors is more challenging. Generally, we have a physically-based "forward model" (denoted as $F$) that relates the geophysical variable we want to observe with our actual measurement; thus, the retrieval problem is essentially deriving the inverse $F$ to map from the observation to our desired geophysical variables. However, deriving the geophysical variables of interest from observations made by passive remote

sensors often is a mathematically ill-posed problem; i.e., there are often many possible values for the geophysical variables that would map through $F$ to our observation, especially given that there is always uncertainty in the observation. Thus, we use additional a-priori information (i.e., information collected before the observation is made) to constrain the retrieval. This is not a new endeavor: scientists have been retrieving information from passive remote sensors for many decades (e.g., Smith et al. 1970), with the mathematical development of these "inverse methods" preceding it (e.g., Twomey 1966). A good high-

level overview of different retrieval methods is given by Maahn et al. (2020), but there exists a large number of detailed texts that explore the retrieval, or inverse theory (e.g., Tarantola 2005; Rodgers 2000).

The challenge with retrievals from passive remote sensors is not only understanding the uncertainty in the retrieval, but also the information content that is offered by the observation itself given that there is also some contribution from the a-priori constraint. Westwater and Strand (1968) provide a concise definition for information content vis-à-vis retrievals: "The

information content…is defined as a reduction in the uncertainty in the (retrieval) after the (observations) are introduced." The information content of a remote sensing measurement is not a new concept, but is an important one as it allows the user of the retrieval to understand how many independent pieces of information are in the observation itself and how that information is distributed among the geophysical variables that are retrieved.

Often, geophysical variables retrieved from remote sensors are used to derive estimates of other geophysical variables. An

example of this is deriving convective available potential energy (CAPE) from a ground-based passive remote sensor from which thermodynamic profiles are retrieved. Blumberg et al. (2017) demonstrated how to use Monte Carlo sampling of the posterior covariance matrix of the retrieved profile to estimate a large number of profiles that would technically satisfy the radiance observation, computed CAPE from each profile, and then estimated the uncertainty in CAPE by looking at the distribution of the CAPE values provided by the Monte Carlo sampling. Monte Carlo sampling is a computational approach





to estimate uncertainties, but a more traditional error propagation approach could have also been adopted (e.g., the "error analysis" chapter in Bevington and Robinson 2003).

However, there are times when geophysical variables are derived using multiple remote sensors, each with their own uncertainties and information contents. In this paper, we explore the idea of propagating uncertainties and information content from multiple instruments through the derivation equation to provide uncertainties and information content of the

derived quantity. To our knowledge, this is the first time this has been demonstrated for information content.

We chose the recent work by Wagner et al. (2022) that derives the profiles of horizontal water vapor and temperature advection using a network of ground-based instruments to illustrate this approach. This paper will first explain our method to propagate the information content, perform a detailed examination of a single case, and then provide a more statistical description of the information content in the derived advection products.

**2. Observations used to compute advection**

Horizontal advection occurs when there is a spatial gradient in a scalar variable over and upwind of desired region, which is then advected over the region. Wagner et al. (2022) used the network of profiling ground-based remote sensors at the Department of Energy's Atmospheric Radiation Measurement (ARM; Turner and Ellingson 2016) Southern Great Plains (SGP; Sisterson et al. 2016) site in north-central Oklahoma. At each of the network sites, there are two instruments: an

Atmospheric Emitted Radiance Interferometer (AERI; Knuteson et al. 2004) and a Doppler wind lidar (DL; Pearson et al. 2009). The AERI is a passive infrared spectrometer, and thus the "TROPoe" algorithm is used to retrieve thermodynamic profiles above the instrument (Turner and Löhnert 2014; Turner and Blumberg 2019). The DL is an active remote sensor that measures radial velocities along the direction of the outgoing laser beam, and by scanning the lidar in a velocity azimuth display (VAD; Browning and Wexler 1968) manner (i.e., making measurements at a number of different azimuth directions

at a constant elevation angle), profiles of horizontal winds can be derived (e.g., Newsom et al. 2017). Key to this study is that we have a full error covariance matrix for all data used in the analysis. We will first describe the two datasets, then discuss how advection is derived from them.

**2.1 Temperature, humidity, and wind retrievals**

The thermodynamic and wind profiles were both retrieved using physical-iterative retrieval methods that are based upon Gaussian statistics; this is usually referred to colloquially as 'optimal estimation'. Optimal estimation approaches provide an error covariance matrix, denoted $S_x$, which embodies the uncertainty for each retrieval. In this study, temperature and humidity retrievals use the TROPoe algorithm, which was explained in Turner and Löhnert (2014). Two separate studies have reframed the derivation of the horizontal winds from VAD scans using a retrieval approach, thereby constraining the

derived winds with a-priori information (Baidar et al. 2023; Gebauer and Bell 2024). We elected to use the Baidar et al.



(2023) wind retrievals, which we will refer to as "DLoe." While the Gebauer and Bell algorithm allows for the inclusion of non-Doppler lidar observations into the wind profile retrieval, in the absence of those additional observations the approach essentially defaults to that of Baidar et al. Therefore, we do not expect the results shown here to have any significant dependence on which DL retrieval algorithm was used.

Following the nomenclature of Rodgers (2000), we will represent the covariance of the a-priori information as $S_a$, the uncertainty in the observations as the covariance matrix $S_m$, and the sensitivity of the forward model $F$ as $K = \partial F / \partial x$, where $x$ is the geophysical variable we are retrieving. The posterior covariance matrix of the retrieval of $x$ is then

$$S_x = (K^T S_m^{-1} K + S_a^{-1})^{-1} \qquad\qquad \text{Eq 1}$$

as given by Equation 3.31 in Rodgers (2000). Most practitioners only show the square root of the diagonal of $S_x$, as this

represents the 1-σ uncertainties at that level for that variable; however, the off-diagonal elements of $S_x$ represent the covariance in the uncertainties between levels and/or variables, and will be important for this study.

The averaging kernel of the retrieval provides a wealth of information about the retrieved quantities. Again, following equation 3.28 in Rodgers (2000), the averaging kernel of the retrieval is computed as

$$A = S_x K^T S_m^{-1} K \qquad\qquad \text{Eq 2}$$

The diagonal of the $A$ is extremely important, as it provides a measure of the degrees of freedom for signal (DFS) that the observations provide to the retrieval for each variable (and as we are working with profiles here, each element of the diagonal is for a specific height above the ground), with the sum of the diagonal (i.e., the trace of A) being the total DFS for the entire retrieval. The DFS at a given height ranges from 0 (i.e., there is no information in the observations) to 1 (i.e., there is perfect information content in the observations). The latter implies that, if there was a perturbation to the state vector (i.e.,

the true atmospheric values of the variables we are retrieving), then the retrieval would perfectly capture the magnitude of that perturbation.

The AERI instruments have diminishing information content on the thermodynamic profile above 1 km, with very little information above 3 km. However, as illustrated in Turner and Blumberg (2019), observations from other instruments can be added as part of the observation vector to improve the retrieved solution (and thus increase its information content). The

TROPoe retrievals used for this work, which were processed by the ARM data center, included AERI radiance data, microwave radiometer brightness temperatures at 23.8 and 31.4 GHz, surface met observations of temperature and relative humidity, and temporally interpolated profiles of temperature and water vapor mixing ratio from the radiosondes launched roughly every 6 hours at the SGP C1 facility as part of the observation vector. However, to prevent overfitting to the C1 radiosondes, the uncertainties in the radiosonde temperature profiles were assumed to be 20 C and 5 g kg⁻¹ at the surface,

decreasing linearly to 4.5 C and 1 g kg⁻¹ at 3 km, to allow the algorithm to place more emphasis on the high-time resolution remote sensors as part of the retrieved profile. The uncertainties in the collocated microwave radiometer brightness temperatures were assumed to be 0.3 K for each frequency. Thus, virtually all of the information content in the TROPoe retrievals below 1.5 km is from the AERI instruments, but the MWR and radiosondes start to have more influence above 2 km (especially on the retrieved water vapor profile).



For this study, we will only use the "down" triangle (Fig 1) discussed in Wagner et al. (2022); namely, the triangle that is created by the site near Waukomis, OK (E37; located at 36.311 N, 97.928 W), the central facility (C1; located at 36.606 N, 97.485 W), and the site near Morrison, OK (E39; located at 36.374 N, 97.069 W).  Note that the distances from E37 to C1, C1 to E39, and E39 to E37 are approximately 50, 45, and 78 km, respectively.

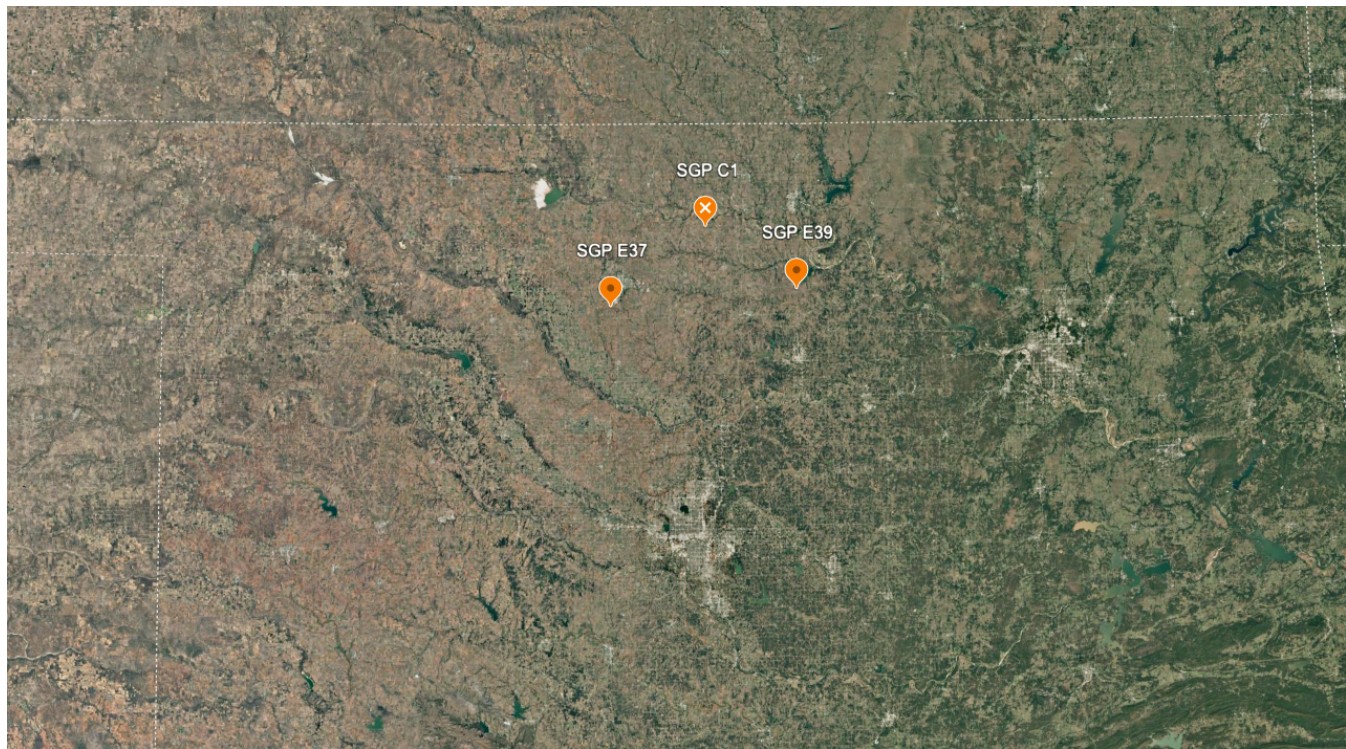

*Figure 1: A map of central to northcentral Oklahoma, showing the locations of the three sites used in this analysis. Extracted from © Google Earth.*

Figure 2 provides an example of the retrieved temperature ($T$; panels a1, a2, a3) and water vapor ($q$; panels b1, b2, and b3) profiles over the SGP site for 13 June 2019 at the E37, C1, and E39 sites.  Similarly, the DLoe-retrieved winds are shown in Fig 2, with $u$ (panels c1, c2, and c3) and $v$ (panels d1, d2, and d3) winds from the three sites, respectively. The thermodynamic and kinematic evolution looks qualitatively very similar across the three sites over this day; however, there are small variations in the retrieved profiles that affect the calculated advection (shown in the next subsection).

Note that for TROPoe, the algorithm simultaneously retrieves both $T$ and $q$ (i.e., $x = [T, q]^T$) so that the posterior covariance matrix $S_{tq}$ (computed using Eq 1) includes the level-to-level covariances of temperature to temperature, water vapor to water vapor, and temperature to water vapor.  Similarly, the DLoe algorithm simultaneously retrieves the $u$ and $v$ wind components (i.e., $x = [u, v]^T$), and thus its posterior covariance matrix $S_{uv}$ includes the cross-correlated errors between $u$ and $v$. For TROPoe, $S_m$ was approximated as a diagonal matrix from the AERI radiance uncertainties (Turner and Blumberg 2019).  For





DLoe, $S_m$ was specified as a diagonal matrix based upon the DL's signal-to-noise ratio at each height (Baidar et al. 2023). The uncertainties in the retrieved $T$ and $q$, and $u$ and $v$ winds, were derived from the square root of the diagonals of $S_{tq}$ and $S_{uv}$ respectively, and are shown in Fig 3. One notable feature is that the uncertainties in the retrieved winds from the E37 DL (Fig 3, panels c1 and d1) are larger than for the other two DLs (Fig 3, panels c2, c3, d2, d3), especially above 1 km.

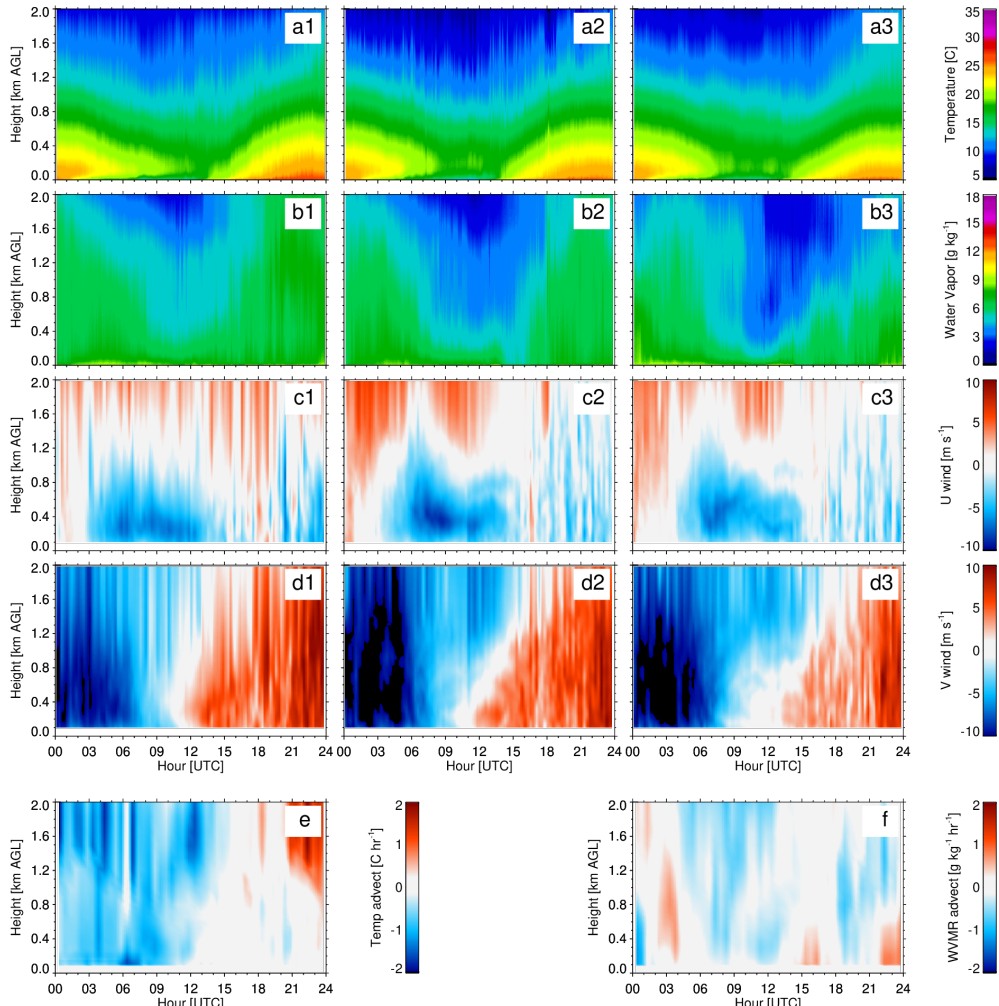

*Figure 2: Time-height cross-sections of temperature and humidity retrieved from the AERIs (rows a and b), and u and v winds retrieved from the DLs (rows c and d), at the E37 (column 1), C1 (column 2), and E39 (column 3) sites for 13 June 2019. The derived temperature advection and water vapor advection fields are in panels e and f, respectively.*



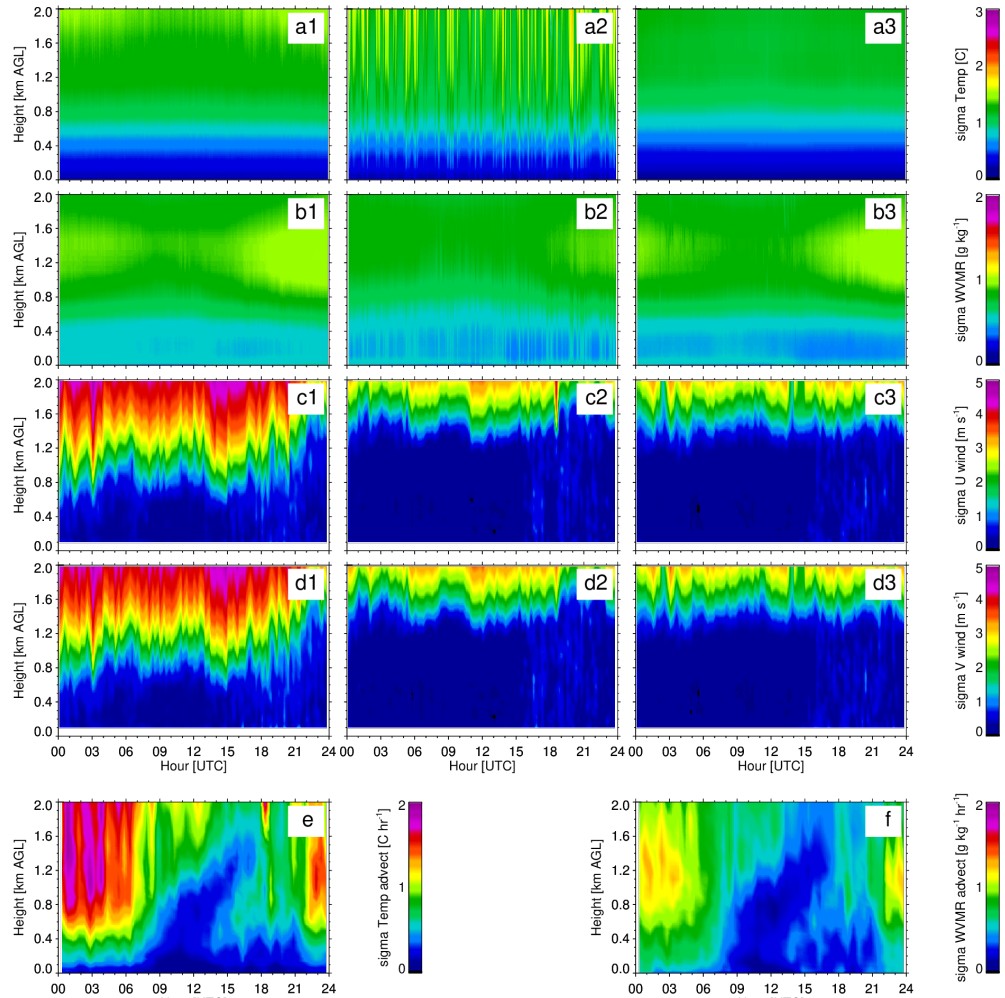

*Figure 3: Same as Fig 2, but showing the 1-σ uncertainties of the various products, which were computed as the square root of the diagonal of the posterior covariance matrices.*


## 2.2 Advection

Michael (1994) demonstrated that the horizontal advection of a scalar $\varphi$ ($\overleftrightarrow{\varphi}$) is computed as a line integral around the network of observations that outlines a polygon in space. Wagner et al. (2022) extended this to vertical remote sensors to get profiles of horizontal advection. For this work here, we used observations at three sites (the triangle shown in Fig 1), and the

advection was computed as

$$\overleftrightarrow{\varphi} = \frac{-\sum_1^3 \overline{\varphi}(\overline{u}\Delta y - \overline{v}\Delta x)}{A_{triangle}} \qquad \text{Eq 3}$$



where $A_{triangle}$ is the area of the triangle (in m²), the summation is over each leg of the triangle, the averaged quantities ($\bar{\varphi}$, $\bar{u}$, and $\bar{v}$) are computed from the two vertices that make up that leg, and $\Delta x, \Delta y$ are the distances between the two vertices of that leg in the zonal and meridional directions (in m) (Wagner et al. 2022). Using Eq 3 as the forward model $F$, we compute the profiles of advection of $\overleftrightarrow{T}$ and $\overleftrightarrow{q}$ simultaneously as

$$\overleftrightarrow{x} = F(x_{E37}, x_{C1}, x_{E39}, w_{E37}, w_{C1}, w_{E39}) \qquad \text{Eq 4}$$

where $\overleftrightarrow{x} = \begin{pmatrix} \overleftrightarrow{T} \\ \overleftrightarrow{q} \end{pmatrix}$, $x = \begin{pmatrix} T \\ q \end{pmatrix}$, and $w = \begin{pmatrix} u \\ v \end{pmatrix}$, where $T$, $q$, $u$, and $v$ are all profiles that have been interpolated to the same vertical grid (as defined by the TROPoe retrievals of $T$ and $q$).

Temperature and water vapor advection fields calculated for 13 June 2019 are shown in Fig. 2 (panels e and f, respectively). Perhaps the most notable feature seen in this example is the deep layer of cold air advection that ends 1200 UTC (Fig 2e) when the meridional wind direction changes from northerly to southerly (Fig 2 panels d1, d2, and 3), which corresponds nicely with the change in the synoptic pattern (Fig 4). Also, there are small pulses of positive water vapor advection at 0300 and 1500 UTC (Fig 2f) that are associated with the onset and demise of the easterly component of the wind (seen in Fig 2 panels c1, c2, and c3).

The advantage of this formulation (Eq 4) is that the uncertainties in the advection of $T$ and $q$ (i.e., $\sigma_{\overleftrightarrow{x}}$) can be easily estimated using standard error propagation techniques (e.g., Bevington and Robinson 2003) as

$$\sigma_{\overleftrightarrow{x}}^2 = \sigma_{x_{E37}}^2 \left(\frac{\partial F}{\partial x_{E37}}\right)^2 + \sigma_{x_{C1}}^2 \left(\frac{\partial F}{\partial x_{C1}}\right)^2 + \sigma_{x_{E39}}^2 \left(\frac{\partial F}{\partial x_{E39}}\right)^2 +$$
$$\sigma_{w_{E37}}^2 \left(\frac{\partial F}{\partial w_{E37}}\right)^2 + \sigma_{w_{C1}}^2 \left(\frac{\partial F}{\partial w_{c1}}\right)^2 + \sigma_{w_{E39}}^2 \left(\frac{\partial F}{\partial w_{E39}}\right)^2 \qquad \text{Eq 5}$$

since there are no correlated errors between any of the six different instruments. Writing this in terms of covariance matrices, we get

$$S_{\overleftrightarrow{x}} = K_{x,E37}^T S_{x,E37} K_{x,E37} + K_{x,C1}^T S_{x,C1} K_{x,C1} + K_{x,E39}^T S_{x,E39} K_{x,E39} +$$
$$K_{w,E37}^T S_{w,E37} K_{w,E37} + K_{w,C1}^T S_{w,C1} K_{w,C1} + K_{w,E39}^T S_{w,E39} K_{w,E39} \qquad \text{Eq 6}$$

where $K$ is the Jacobian of $F$ for both $x$ and $w$, computed at the three different sites, and the superscript $T$ in this context represents matrix transpose.

Using this approach, the uncertainties in the retrieved thermodynamic and wind profiles were propagated to provide the uncertainties in the temperature and moisture advection for the observations on 13 June 2019. The temperature retrievals show very small uncertainties, less than 0.5 C, below 500 m but increase to nearly 1.5 C near 2 km above ground level (Fig 3 panels a1, a2, and a3). High frequency variation is seen in the uncertainties for the C1 temperature retrieval (Fig 3 panel a2), which is a result of temporal variation in the uncertainties in the downwelling infrared radiation observations in the temperature sensitive spectral region observed by the AERI at that site. The water vapor uncertainty data is qualitatively very similar across the three sites, with again the lowest uncertainties below 500 m (Fig 3 panels b1, b2, and b3). The



uncertainties in the *u* and *v* winds from the DLs are relatively low below about 1.5 km, but above that height the
uncertainties increase drastically due to the poor signal-to-noise ratio in the DL radial velocity observations above 1.5 km

190    due to a relative lack of aerosols. However, Fig 3 (panels c1 and d1) also demonstrates that the DL at the E37 site has much
poorer data quality (i.e., higher instrument noise levels) relative to the DLs at the other two sites (Fig 3 panels c2, c3, d2, and
d3). It is important to note here that the a-priori information used for TROPoe and DLoe was identical at the three sites;
thus, the variability across the three sites in the derived uncertainties and information is due to the differences in the
instrument uncertainties at the different locations.

195    We propagated the uncertainties of the three AERIs and three DLs (i.e., the instrument-specific posterior covariance
matrices) to derive the covariance of the advection (i.e., $S_{\bar{x}}$) using Eq 6. Time-height cross-sections of the 1-$\sigma$ uncertainties
of the temperature and moisture advection are shown in Fig 3, panels e and f, respectively. The uncertainties in the both the
temperature and water vapor advection for this day (13 June 2019) are small near the surface, and generally increase with
height. In particular, the uncertainty in the temperature advection above 1.2 km from 0000 to 0700 UTC is quite large,

200    suggesting that the cold air advection shown in this time period (Fig 2e) has a lot of uncertainty. However, there are
particularly low uncertainties in the derived advection from approximately 0900 to 1500 UTC from the surface to nearly
1500 m that seem associated with the change in the synoptic pattern.

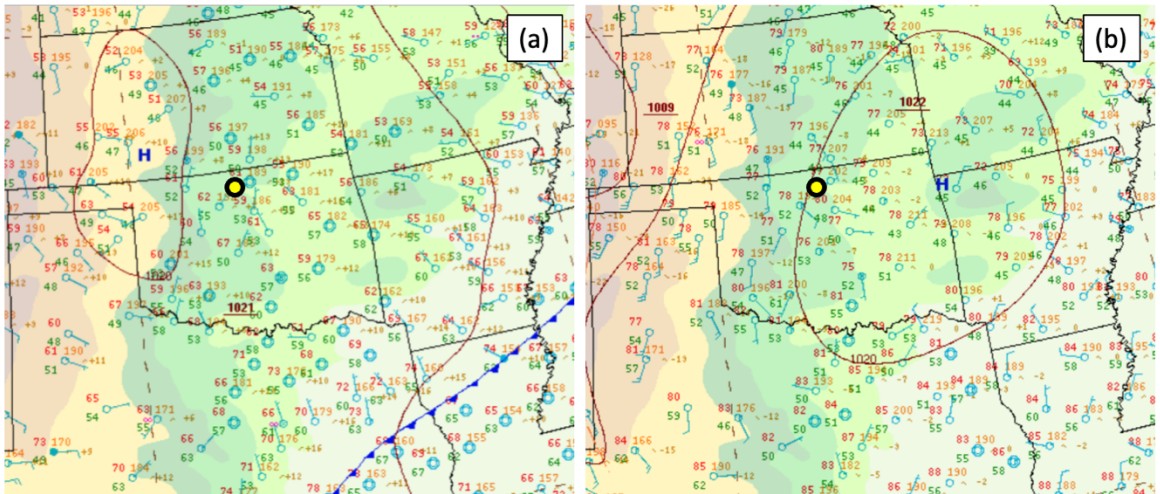

*Fig 4: Surface synoptic maps at 0600 (a) and 1800 (b) on 13 June 2019. The yellow dot indicates the*
*approximate location of the ARM C1 site. From the NOAA Weather Prediction Center map archive at*
*https://www.wpc.ncep.noaa.gov/archives/web_pages/sfc/sfc_archive.php*





## 3. Information content propagation


For retrievals, the concept of information content is used to express how much information we are gaining through the measurements over our prior knowledge. Information content heavily depends on the instrument's noise level and the sensitivity of the forward model. It is known that the information content in the AERI thermodynamic retrievals is not large and that it decreases rapidly with height above the surface (e.g., Turner and Löhnert 2014; Turner and Löhnert 2021), with

approximately a total of 5 for $T(z)$ and between 3-to-5 for $q(z)$ below 3 km. The DL generally has high information content (near unity for each height level) where there is sufficient aerosol concentration to provide the backscattered signal (which is usually only within the atmospheric boundary layer). The question we want to address here is how does the information content in individual instruments translate to information content for the derived advection that require using data from multiple instruments?

For this, we need to look at the averaging kernel in more detail. The averaging kernel $A$ (Eq 2) can also be expressed as:

$$A = \frac{K^T S_m^{-1} K}{K^T S_m^{-1} K + S_a^{-1}}$$

$$= \frac{K^T S_m^{-1} K + S_a^{-1} - S_a^{-1}}{K^T S_m^{-1} K + S_a^{-1}}$$

$$= I - \frac{S_a^{-1}}{K^T S_m^{-1} K + S_a^{-1}}$$

$$= I - \frac{S_x}{S_a} \qquad\qquad\qquad \text{Eq 7}$$

This was also shown by Cadeddu et al. (2017), where $I$ is the identity matrix. The prior covariance matrix $S_a$ illustrates the climatological "volume" of state vector, and posterior covariance matrix $S_x$ is the "volume" of that space that results after the retrieval is performed and the information from the observations are included. If there is no information about the state vector from the observations, then $S_x$ will be approximately $S_a$, and thus $A$ is approximately 0. If there is a lot of information in the observations, then $S_x$ will be markedly smaller than $S_a$ and thus $A$ will be approximately 1. This matches the high-level

definition of information content provided by Westwater and Strand (1968) well.

From Eq 7, we see that if we can obtain $S_{a,\bar{x}}$ (the advection prior covariance) and $S_{\bar{x}}$ (the advection posterior covariance) then we can estimate the information content of the derived advection that we are gaining through the measurements. We have already demonstrated that $S_{\bar{x}}$ can be obtained through error propagation using Eq 6, since the AERI and DL retrievals provide posterior covariance matrices. To get $S_{a,\bar{x}}$, we again use Eq 6, but instead replace the posterior covariance matrices

$S_x$ and $S_w$ with the prior covariance matrices used in the TROPoe ($S_{a,x}$) and DLoe ($S_{a,w}$) retrievals, respectively. Note that the three facilities (E37, C1, and E39) use identically the same prior covariance matrix in TROPoe, and the same is true for DLoe; these have been derived using 20 years of summertime radiosonde data launched at the SGP central facility.



## 4. Example

To compute the information content (DFS) for the temperature and water vapor advection, the prior and posterior covariance
matrices from the observations were propagated through the forward model using Eq 6, and then the averaging kernel $A$ was
computed using Eq 7. The diagonal of $A$ provides a profile of DFS for both temperature and water vapor advection, which
are shown in Fig 5 (panels e and f, respectively). There is a striking similarity to the spatial patterns of the 1-σ uncertainties
(Figs 3e and 3f) and the DFS (Figs 5e and 5f) time-height cross-sections. Generally speaking, the DFS and 1-σ uncertainties
are anti-correlated, with higher DFS values being associated with lower 1-σ uncertainties. In the region of cold air advection
above 1.2 km from 0000 to 0700 UTC, the DFS is very small for temperature advection (Fig 5e) suggesting that there is no
information in that region and thus those advection values should not be trusted. However, the DFS figures also suggest that
the information content on advection can often be near unity at heights approaching 2 km (Fig 5, panels e and f), even
though the AERI's information content is very limited with DFS < 0.05 at all levels above 50 m for both $T$ and $q$ (Fig 5,
panels a1-a3 and b1-b3). The DL at the E37 site also is clearly the outlier of the three DLs from in information content
perspective, which can be seen by comparing the c1 and d1 panels with the c2, c3, d2, and d3 panels in Fig 5.

There are several natural questions that could be asked to better understand the information content results. For example, are
the site-to-site differences in the posterior covariances (as evidenced by the changes in the 1-σ uncertainty profiles shown in
Fig 5 for a given variable like T or u) impactful on the advection DFS profiles? To test this, we performed two tests: use the
posterior covariance matrix from the C1 retrieval as the posterior for the E37 and E39 retrievals for both (a) $T$ and $q$, and (b)
$u$ and $v$. In both of these tests, there was relatively little change to the resulting time-height profile of DFS for temperature
and moisture advection (DFS differences were less than 0.1, not shown). We found this surprising, especially since the E37
DL has much larger uncertainties in $u$ and $v$ above 1 km than the other two sites; however, if we used the E37 DL posterior
for all three sites, then the information content decreased nearly 0.1 uniformly above 1.2 km (not shown). Another sensitivity
test performed was to inflate the TROPoe posterior covariances by a factor of 2. The new DFS time-height cross-section for
the temperature and advection data is shown in Fig 6 (panels a and b, respectively). Comparing these DFS results with the
baseline (Fig 5, panels e and f) shows a qualitatively similar evolution of the DFS profiles, but also a decrease in the DFS for
temperature and moisture advection by approximately 0.2 to 0.4 along the top contour (Fig 6, panels c and d). Interestingly,
inflating the DLoe posterior covariance matrices by a factor of 2 had little effect (DFS differences less than 0.1) below 2 km,
with some DFS differences approaching 0.3 around 2.5 km (not shown).



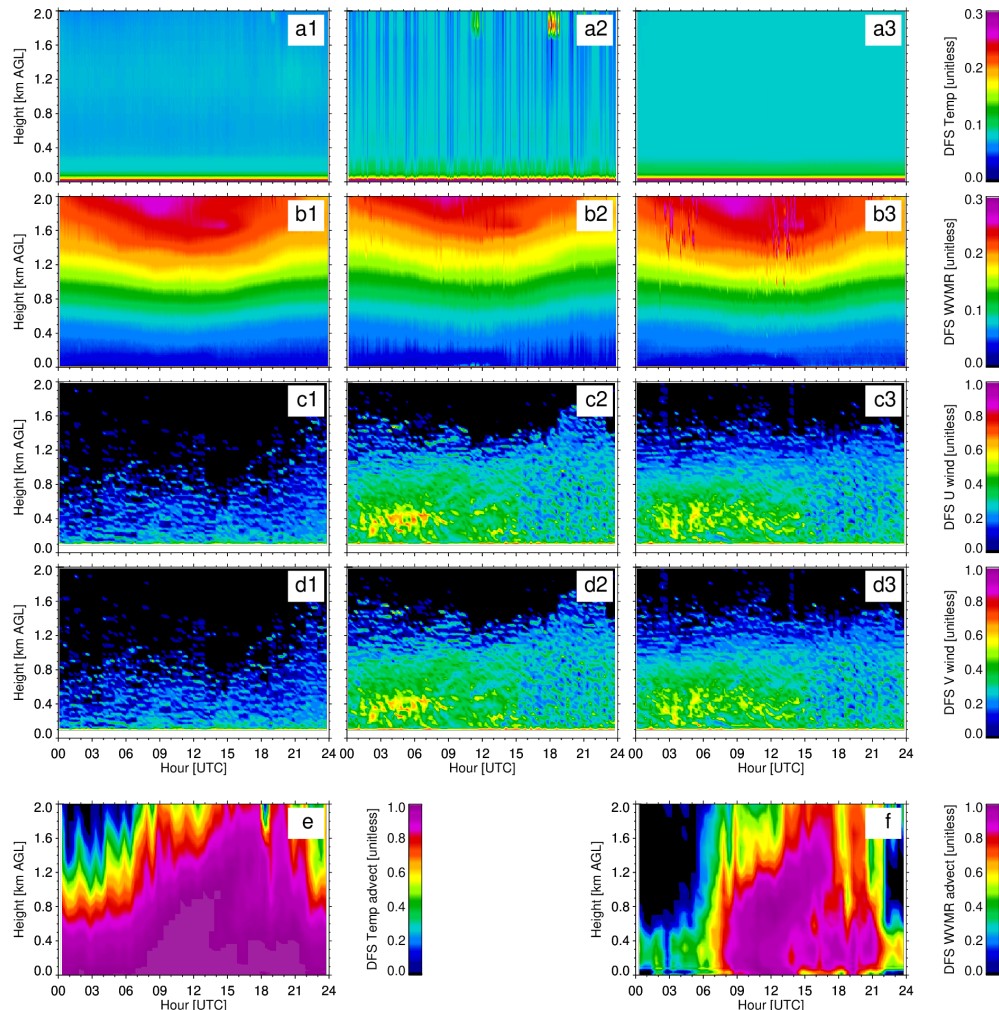

*Figure 5: Same as Fig 2, but showing the DFS of the various products.*

So far, we have focused on the diagonals of the covariance matrices, as the square root of the diagonal provides the 1-σ
profile of uncertainties. Figure 7 shows the 1-σ profiles derived from the advection's prior covariance (i.e., $S_{a,\bar{x}}$) and the
mean 1-σ profile from the advection's posterior covariance (i.e., from $S_{\bar{x}}$) for the 13 June 2019 case. Clearly, the addition of
the observations is adding information, as the posterior uncertainty profile is smaller. However, it is important to realize that
the information content profile is not 1 minus the ratio of these two profiles; instead, as illustrated in Eq 8, the off-diagonal
elements also play a role.



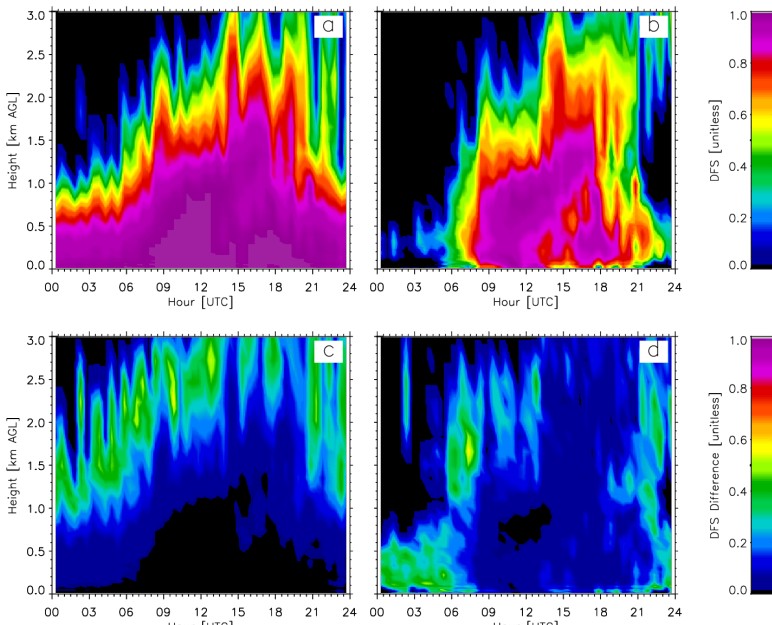

*Fig 6: The time-height cross-section of DFS for temperature and moisture advection (panels a and b, respectively) for 13 June 2019, where the posterior covariance matrices from the TROPoe retrievals of T and q were inflated by a factor of 2. Panels c and d show the time-height cross-section of the differences of the DFS from Fig 5 (panels e and f) with panels a and b here.*

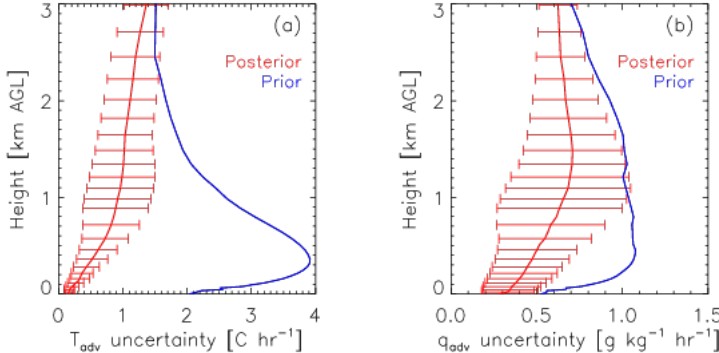

*Fig 7: The 1-σ profiles for temperature (a) and water vapor (b) advection, derived from the prior (blue) and posterior (red) covariance matrices for 13 June 2019. The posterior profile is the mean at each height over the day, with the error bars showing the standard deviation at each height over that day.*



The off-diagonal elements from both $S_{a,\vec{x}}$ and $S_{\vec{x}}$ are shown in Fig 8, where the covariance matrices were converted to correlation matrices for display purposes. Since these covariance matrices are symmetric, only half of each is shown with the prior shown below the diagonal and the posterior above the diagonal. Note the large magnitude of the level-to-level correlation in the advection of temperature with itself (Fig 8a), water vapor with itself (Fig 8b), and the cross-correlation of

temperature and water vapor advection (Fig 8c). However, after the retrieval, both the magnitude of the diagonal (Fig 7) and the magnitude of the off-diagonal terms are markedly reduced. There are some negative correlations seen between two different levels (e.g., the correlation of temperature advection with itself at 300 and 900 m is approximately -0.3 – see Fig 8a); these features are similar to the structure of the correlation matrices in the TROPoe covariance matrices but much weaker (see Turner and Blumberg 2019, Fig 10 for an example).


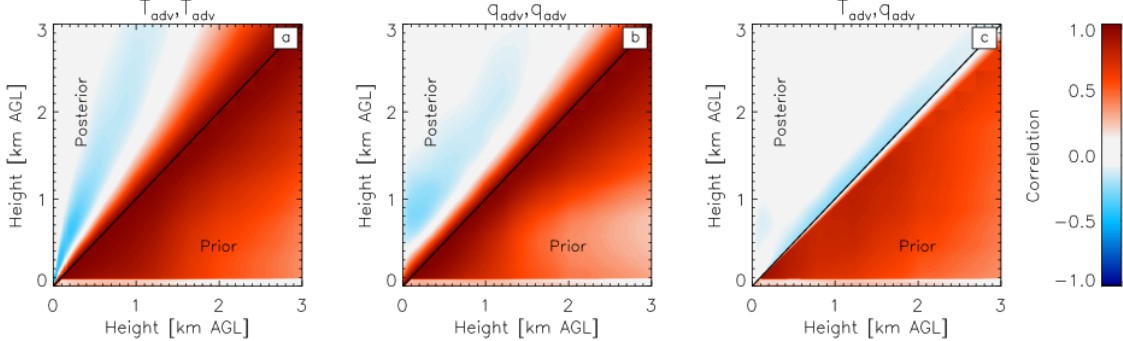

*Fig 8: The level-to-level correlations between the advection of temperature/temperature (a), water vapor/water vapor (b), and temperature/water vapor (c) for 13 June 2019. The prior correlations are shown below the diagonal, and the posterior correlations are above the diagonal.*

## 5. Statistical summary

The example on 13 June 2019 shown in Figs 2, 3, and 5 provides an illustration of the derived advection profiles, its uncertainties, and its information content, which uses retrievals from 6 independent instruments in the derivation. This

particular example was chosen because there was marked temporal variability in the 24-hour period. However, we are interested in more general statements about the uncertainty and information content in the derived advection. Recently, nearly 2 years of advection data were derived from the SGP observations (Jan 2018 to Sep 2019). This advection dataset is characterized as a function of synoptic pattern in Wagner et al. (2024); however, that analysis did not include a description of the information content. Here, we analyze that same dataset to provide a sense of the average magnitude of the information

content in both the temperature and water vapor advection, how that information content varies both over the diurnal cycle



and as a function of height, over two 4-month periods. Table 1 indicates the number of days of data that were available for each month in the 2-year record, and the number of cases in the "cool" and "warm" seasons.

*Table 1: Number of days per month between 14 Jan 2018 and 17 Sep 2019 used in the analysis for the two seasons*

| "Cool Season" | | "Warm Season" | |
|---|---|---|---|
| **Month** | **Number of Days** | **Month** | **Number of Days** |
| February | 27 | June | 27 |
| March | 54 | July | 31 |
| April | 28 | August | 25 |
| May | 43 | September | 15 |
| **Total** | **152** | **Total** | **98** |

Time-height cross-sections of the mean information content for temperature advection and water vapor advection for the cool season are shown in the top row of Fig 9. Note that the TROPoe algorithm has a minimum boundary layer height of 300 m; thus, the nocturnal boundary layer heights are largely this value. The standard deviation of the DFS data for each time and height are shown in the bottom row of Fig 9, and provide a measure of the variability in the DFS across the nearly 150 days in this cool season analysis.

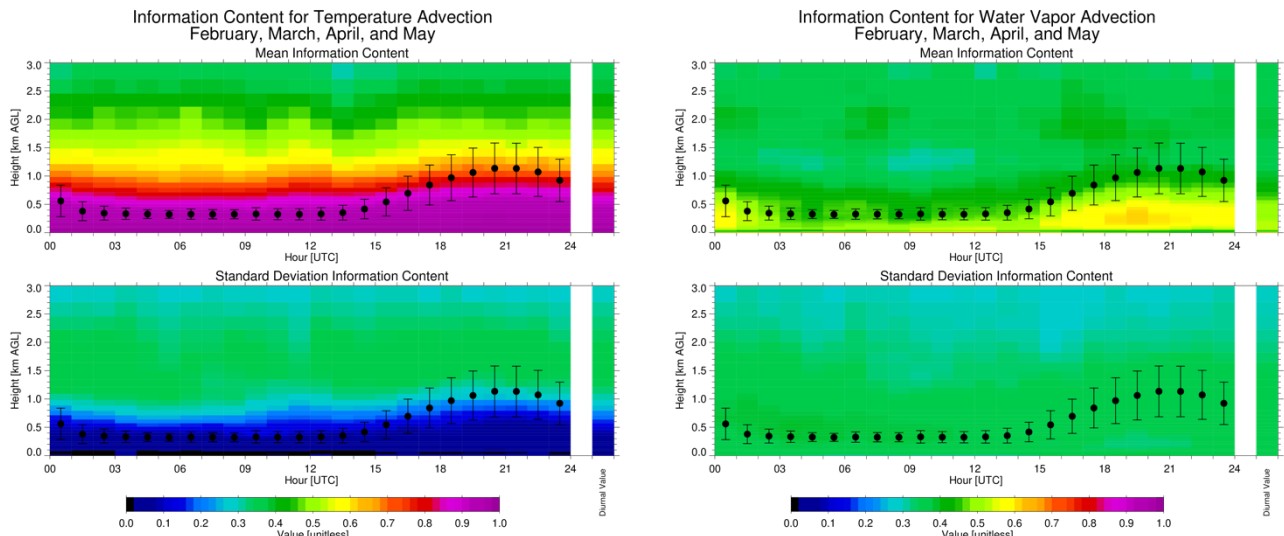

*Figure 9: The mean diurnal information content (top row) for temperature advection (left) and water vapor advection (right) for the "cool season" (table 1). The variability of the information content, represented as the standard deviation, over this four-month period is shown in the bottom row. The black dots illustrate the mean depth of the atmospheric*



*boundary layer, which is derived from the TROPoe retrievals using a parcel method (e.g., similar to that used in Nielsen-Gammon et al. 2008), with the error bars indicating the variability of this height for each time over the days included in the analysis. The single bar on the right side of all four panels shows the value averaged over the entire diurnal cycle.*


As can be seen, the information content for temperature advection (Fig 9, left column) is above 0.9 (with a standard deviation less than 0.15) for most heights below 700 m, decreasing to mean value 0.8 by 1000 m. The standard deviation of the temperature DFS increases to about 0.4 at altitudes above 900 m. The mean DFS in the water vapor (Fig 9, right column) has smaller values, with mean values between 0.5 to 0.7 in the lower-to-middle part of the daytime boundary layer

(i.e., between 1400 to 2400 UTC), with the mean DFS decreasing to 0.4 to 0.5 at and above the top of the boundary layer. The standard deviation of the water vapor advection DFS is approximately 0.3 to 0.4, regardless of time of day or height. Figure 10 shows the same statistics for the warm season. There is little difference in the mean DFS or its standard deviation for the temperature advection over the diurnal cycle or vertically between the cool season (Fig 9, left column) and warm season (Fig 10, left column). However, there is a marked difference in the water vapor information content between the two

seasons. The mean information content is much larger in the daytime boundary layer during the warm season, with mean values of 0.7 to 0.8 in the warm season vs 0.5 to 0.6 in the cool season. The variability in the DFS for water vapor advection is also about 50% smaller in the daytime boundary layer in the warm season vs the cool season. There is also an increase in the mean water vapor advection DFS above 2 km in the warm season, which is primarily contributed by the use of the radiosondes data in the TROPoe retrievals with a smaller contribution from the microwave radiometer brightness

temperature observations, both of which are having a larger impact due to the overall wetter conditions in the warm season.

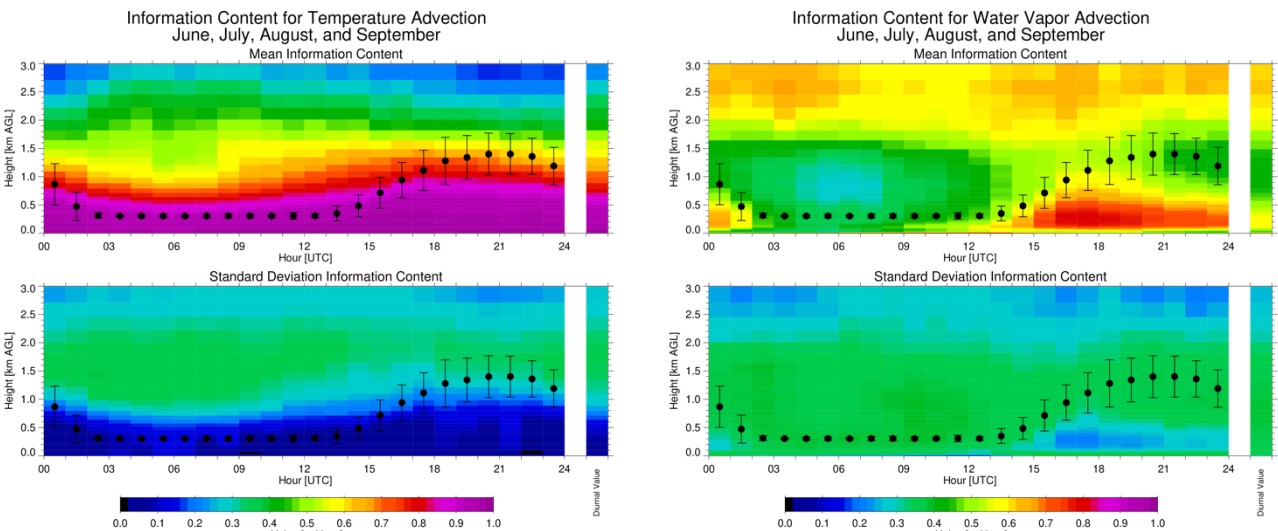

*Figure 10: Same as Fig 8, but for the "warm season" (Table 1).*



## 6. Conclusions

To properly interpret remote sensing observations, which are often constrained using a-priori information, it is important to understand its uncertainties and its information content. Some geophysical variables are derived from remote sensing observations, and thus the uncertainties and information content need to be propagated through the derivation equation. This work demonstrates, for the first time, how to propagate information content from multiple remote sensors through a derivation equation for a new quantity. The key is to propagate the uncertainties from each individual retrieval through the derivation equation simultaneously, then propagate the uncertainties from the a-priori constraints through the same equation in the same manner, and look at the ratio of the covariance matrices derived from the two datasets.

We illustrated this approach using a network of remote sensors to derive the horizonal advection. To derive advection, we needed to have (at least) three non-colinear sites that measure profiles of the quantity of interest (in this case, temperature and humidity) with wind profiling at the same locations. In our case, we had six separate instruments, with three providing thermodynamic profiles (using the same a-priori information) and three providing kinematic profiles (again, using the same a-priori information for all three). However, because of differences in the noise characteristics of the different instruments, the uncertainties and information content derived from each individual instrument varied. The posterior covariance matrices (i.e., the individual retrieval uncertainties) were used to derive uncertainties and information content profiles for the derived temperature advection and water vapor advection profiles.

A statistical analysis of the information content profiles demonstrates that there is nearly perfect information content (i.e., DFS close to 1) for temperature advection below 700 m. This suggests that if there is a true change in the temperature advection below that level that the observed temperature advection would capture the magnitude that change at the right level. The associated information content for water vapor advection is different though; it is a strong function height, time-of-day, and season. Nonetheless, the daytime mean information content for water vapor advection in the boundary layer in the warm season is above 0.7, suggesting that the magnitude any true perturbation in the water vapor advection would be largely captured by this instrument suite.

This work demonstrates how to derive the information content of an observation that is derived from multiple datasets that were derived from remote sensors. The key aspect is to frame the individual derivations as retrievals, so that both prior and posterior covariance matrices are available. Propagating information content, as was illustrated here, can inform the user of the derived data on where the signal-to-noise is the best, and potentially reduce the opportunity to misuse the data.

### Code availability.

The analysis code used in this work was written in IDL, which is available from the corresponding author upon request.



**Data Availability Statement.**

The data used in this research effort are available via the ARM data archive: www.arm.gov.

**Author Contributions.**

DDT conceived of this project, with critical input from MPC; JS and TJW helped to refine the concept regarding its
application to advection. DDT developed the analysis code, and all coauthors evaluated and discussed the results. The
manuscript was written by DDT with contributions from all coauthors.

**Competing Interests**

The authors declare that they have no competing interests.

**Acknowledgments.**

This work was supported primarily by the NOAA Global Systems Laboratory and U.S. Department of Energy's
Atmospheric System Research (ASR) program, which is within the Office of Science Biological and Environmental
Research division, under grants 89243023SSC000117 and DE-SC0024048. MC is supported by the U.S. Department of
Energy, Office of Science, Office of Biological and Environmental Research, Atmospheric Radiation Measurement
Infrastructure, under Contract DE-AC02-06CH11357. Data were obtained from the Atmospheric Radiation Measurement
(ARM) facility, a U.S. Department of Energy (DOE) Office of Science user facility managed by the Office of Biological and
Environmental Research.

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
