# Peer review of "Propagating Information Content: An Example with Advection"

_EGUsphere, 2024_

## Referee Comment (RC2)

**Review of the Manuscript: "Propagating Information Content: An Example with Advection"**

The present paper compares the propagated uncertainties retrieved with observations and with a priori data; in order to account for the information content of the derived geophysical variables. In particular, this is done to propagate the uncertainties of water vapor and temperature advection estimated from the measurements of six different instruments. The scientific content of this paper is relevant and rigorously assessed and most of the explanations are clear and well structured. Moreover, it does provide a novel quantification of the information content of a variable of crucial interest in atmospheric dynamics, which is advection. And this is of great interest given that the growing tendency of remote sensors around the world has a great potential to estimate this variable given the synergistic use of ground-based sensors and this needs to be carefully evaluated to make sure that a proper quantification of this variable is performed. For all these reasons, my recommendation is to accept the present paper. However I do have some comments that have to be addressed before its publication. I address them in the following lines.

Line 124 and : Add the "°" to 20ºC and 4.5ºC

Lines 130–135 and Figure 1: The location of the three facilities is shown; however it is not very clear from the figure. It reads "*Note that the distances from E37 to C1, C1 to E39, and E39 to E37 are approximately 50, 45, and 78 km, respectively*" But this is not clearly noted by the figure as there is no any length scale in it. Please add one. Moreover, it is a quite big area that is shown surrounding this triangle of facilities and nothing special is shown in that area, so I would suggest either to make more a zoom into the facilities or add the important information that the authors want to show with such a big area. The map is from Google Earth, and the green and brown colors might stand for the vegetation and ground in general but this is not mentioned nor of importance as far as the authors are explaining. So my suggestion for them is to please provide a more accurate map in which the important information is presented and the no necessary features do not occupy most of the area.

Figure 2: This figure already has a lot of valuable information, however I think that it should be mentioned somewhere that it is in time UTC and what is the corresponding local time, because the diurnal cycle is quite visible but it is of course shifted due to the UTC time in the US. Please clarify this either on the figure itself or in its explanation around lines 135–140.

In general I think that the authors have a lack of citations, and although most of the work is already explained with the current references; I think that there are some aspects that would become more clear how the state of the art is if authors provide more information about previous works. For example in line 59 where they say: "*Often, geophysical variables retrieved from remote sensors are used to derive estimates of other geophysical variables.*" And from lines 65 to 70, authors could provide a wider overview of the geophysical variables that are estimated via multiple remote sensors each with their own uncertainty. What other variables have been propagated their uncertainties besides the information content?

Figure 3: fast temporal variations appear to be visible in this figure, please specify if the temporal resolution of these uncertainties is the same one as that of its variables in figure 2. Additionally, the AERI retrievals (with TROPoe) temporal resolution is not specified and it should be. Please add this information, maybe around line 81 where the AERI is introduced.

Lines 196–200: The authors say: " *The uncertainties in the both the temperature and water vapor advection for this day (13 June 2019) are small near the surface, and generally increase with height. In particular, the uncertainty in the temperature advection above 1.2 km from 0000 to 0700 UTC is quite large, suggesting that the cold air advection shown in this time period (Fig 2e) has a lot of uncertainty*." Although the actual values of "small" , "quite large" and " a lot" are seen in Fig. 3, these sentences together sound rather qualitative, so I suggest that the authors put some numbers on this and maybe comment on how small or large are these uncertainties compared with the actual advection values on figure 2.

Figure 4: in its caption specify the hour, it's UTC again. And what do the colors in the map stand for? That should be clear.

Lines 201–203: The authors say "*However, there are particularly low uncertainties in the derived advection from approximately 0900 to 1500 UTC from the surface to nearly 1500 m that seem associated with the change in the synoptic pattern.* " This idea is not fully clear, please elaborate more: What kind of change are you referring to in the synoptic pattern? How does it relates to the advection of temperature and water vapor and subsequently to their uncertainties? Please clarify this.

As a general comment: on section *4 Examples*, the authors carefully assess the information content and, through plotting the degrees of freedom (DFS) they identify the regions (times and heights) for which the uncertainty of the profiles and the subsequent thermal and water vapor advection are smaller and therefore they could be trusted. Moreover, on section 5 *Statistical summary*, the authors explore how this uncertainty assessment can be utilized to investigate temporally larger datasets and how it would change for cool and warm seasons. Additionally, the authors mention more than once (for example in line 334) that a large information content implies that the magnitude of any true perturbation would be captured by the present assessment. However, I think that it would be useful to mention if there has been any attempt to show if this is really the case when comparing the present advection estimation with other assessments of this variable performed with very different techniques, such as models like large eddy simulations or via in situ measurements. This comparison does not necessarily needs to be presented in the current paper, but I think it is an important aspect to consider, given that datasets as large as almost 2 years are considered in the present paper and so far only retrievals from ground-based remote sensors and their uncertainties are considered. Therefore, a mention on these other comparisons would be elucidating.

The conclusion is generally clear and well-structured, so no big changes are suggested for it. However, it seems that it assumes that the current approach is the only one possible to retrieve advection and its uncertainties from ground-based remote sensors, and this is not necessarily the case. Also, regarding the information content, it is true that to the best of our knowledge there is no other work that determines how to propagate information content from multiple remote sensors; but in line 316 it seems to be assumed that this is the one and

only way to do it, when in reality other techniques can also be developed for other remote sensors that may not be exactly the AERI and DWL. This suggestion is not mandatory, but I would recommend to soften some of these sentences. And as examples of previous works deriving advection from remote sensors, please refer to: Schween, Jan H., Susanne Crewell, and Ulrich Löhnert. "Horizontal-humidity gradient from one single-scanning microwave radiometer."

---

## Author Comment (AC1)

*17 April 2025*

*We appreciate the comments raised by the two reviewers, and thank them for the time and energy they spent on our paper.* Their comments below are in black text*, and our replies to each comment are in blue italics. Any* text directly added to the manuscript *will be shown in green italics in our reply below.*

**Reviewer #2:**

The present paper compares the propagated uncertainties retrieved with observations and with a priori data; in order to account for the information content of the derived geophysical variables. In particular, this is done to propagate the uncertainties of water vapor and temperature advection estimated from the measurements of six different instruments. The scientific content of this paper is relevant and rigorously assessed and most of the explanations are clear and well structured. Moreover, it does provide a novel quantification of the information content of a variable of crucial interest in atmospheric dynamics, which is advection. And this is of great interest given that the growing tendency of remote sensors around the world has a great potential to estimate this variable given the synergistic use of ground-based sensors and this needs to be carefully evaluated to make sure that a proper quantification of this variable is performed. For all these reasons, my recommendation is to accept the present paper. However, I do have some comments that have to be addressed before its publication, I address them in the following lines.

*We thank the reviewer for their comments and their time!*

Line 124 and : Add the "°" to 20°C and 4.5°C

*Done*

Lines 130-135 and Figure 1: The location of the three facilities is shown; however it is not very clear from the figure. It reads "Note that the distances from E37 to C1, C1 to E39, and E39 to E37 are approximately 50, 45, and 78 km, respectively" But this is not clearly noted by the figure as there is no any length scale in it. Please add one. Moreover, it is a quite big area that is shown surrounding this triangle of facilities and nothing special is shown in that area, so I would suggest either to make more a zoom into the facilities or add the important information that the authors want to show with such a big area. The map is from Google Earth, and the green and brown colors might stand for the vegetation and ground in general but this is not mentioned nor of importance as far as the authors are explaining. So my suggestion for them is to please

provide a more accurate map in which the important information is presented and the no necessary features do not occupy most of the area.

*Good suggestion. We have markedly improved figure 1 to show both the larger domain, as well as the three sites with distances between them.*

Figure 2: This figure already has a lot of valuable information, however I think that it should be mentioned somewhere that it is in time UTC and what is the corresponding local time, because the diurnal cycle is quite visible but it is of course shifted due to the UTC time in the US. Please clarify this either on the figure itself or in its explanation around lines 135-140.

*We added this to the caption: "The time (x-axis) is UTC; local time is UTC – 5."*

In general I think that the authors have a lack of citations, and although most of the work is already explained with the current references; I think that there are some aspects that would become more clear how the state of the art is if authors provide more information about previous works. For example in line 59 where they say: "Often, geophysical variables retrieved from remote sensors are used to derive estimates of other geophysical variables." And from lines 65 to 70, authors could provide a wider overview of the geophysical variables that are estimated via multiple remote sensors each with their own uncertainty. What other variables have been propagated their uncertainties besides the information content?

*There are many (many) papers that show the derivation of variables from retrievals; for example, CAPE has been derived from space-borne thermodynamic soundings, cloud properties from satellite radiances, etc. However, there are relatively few examples where uncertainties in retrievals have been propagated to provide uncertainties in derived variables. The Blumberg et al. 2017 paper is the only example for which we are aware.*

Figure 3: fast temporal variations appear to be visible in this figure, please specify if the temporal resolution of these uncertainties is the same one as that of its variables in figure 2. Additionally, the AERI retrievals (with TROPoe) temporal resolution is not specified and it should be. Please add this information, maybe around line 81 where the AERI is introduced.

*We added the temporal resolution of the TROPoe and DLoe retrievals to the first paragraph of section 2.1, as suggested.*

Lines 196-200: The authors say: "The uncertainties in the both the temperature and water vapor advection for this day (13 June 2019) are small near the surface, and generally increase with height. In particular, the uncertainty in the temperature advection above 1.2 km from 0000 to 0700 UTC is quite large, suggesting that the cold air advection shown in this time period (Fig 2e) has a lot of uncertainty." Although the actual values of "small", "quite large" and "a lot" are seen in Fig 3, these sentences together sound rather qualitative, so I suggest that the authors

put some numbers on this and maybe comment on how small or large are these uncertainties compared with the actual advection values on figure 2.

*We have added some specificity to the paragraph with "(less than 0.3 K hr$^{-1}$ and 0.5 g kg$^{-1}$ hr$^{-1}$, respectively)" and "(larger than 1.5 K hr$^{-1}$)".*

Figure 4: in its caption specify the hour, it's UTC again. And what do the colors in the map stand for? That should be clear.

*We believe you are referring to Fig 5 here. As the caption of Fig 5 says "Same as Fig 2", which has an explanation of the time, we believe that the time is clear.*

Lines 201-203: The authors say "However, there are particularly low uncertainties in the derived advection from approximately 0900 to 1500 UTC from the surface to nearly 1500 m that seem associated with the change in the synoptic pattern. "This idea is not fully clear, please elaborate more: What kind of change are you referring to in the synoptic pattern? How does it relates to the advection of temperature and water vapor and subsequently to their uncertainties? Please clarify this.

*We have added this phrase to the end of that sentence: "(i.e., the change in direction of the low winds shown in Fig 4)".*

As a general comment: on section 4 *Examples,* the authors carefully assess the information content and, through plotting the degrees of freedom (DFS) they identify the regions (times and heights) for which the uncertainty of the profiles and the subsequent thermal and water vapor advection are smaller and therefore they could be trusted. Moreover, on section 5 Statistical summary, the authors explore how this uncertainty assessment can be utilized to investigate temporally larger datasets and how it would change for cool and warm seasons. Additionally, the authors mention more than once (for example in line 334) that a large information content implies that the magnitude of any true perturbation would be captured by the present assessment. However, I think that it would be useful to mention if there has been any attempt to show if this is really the case when comparing the present advection estimation with other assessments of this variable performed with very different techniques, such as models like large eddy simulations or via in situ measurements. This comparison does not necessarily needs to be presented in the current paper, but I think it is an important aspect to consider, given that datasets as large as almost 2 years are considered and in the present paper and so far only retrievals from ground-based remote sensors and their uncertainties are considered. Therefore, a mention on these other comparisons would be elucidating.

*The original paper on the advection method used here, namely the Wagner et al. 2022 paper, includes some analysis relative to the High-Resolution Rapid Refresh (HRRR) model.*

The conclusion is generally clear and well-structured, so no big changes are suggested for it. However, it seems that it assumes that the current approach is the only one possible to retrieve advection and its uncertainties from ground-based remote sensors, and this is not necessarily the case. Also, regarding the information content, it is true that to the best of our knowledge there is no other work that determines how to propagate information content from multiple remote sensors; but in line 316 it seems to be assumed that this is the one and only way to do it, when in reality other techniques can also be developed for other remote sensors that may not be exactly the AERI and DWL. This suggestion is not mandatory, but I would recommend to soften some of these sentences. And as examples of previous works deriving advection from remote sensors, please refer to: Schween, Crewell, and Löhnert: "Horizontal-humidity gradient from one single-scanning microwave radiometer"

*We have removed the phrase "for the first time" in the first paragraph in the conclusions. Additionally, we have added two new sentences at the end of the paper that hopefully captures this suggestion.* *"However, it has been shown that a single microwave radiometer making azimuth scans can identify spatial gradients of water vapor (Schween et al. 2011). If this was paired an instrument measuring horizontal wind profiles, potentially water vapor advection could be derived, but the propagation of uncertainties and information content could be performed the same way as shown here."*

---

## Author Comment (AC2)

*17 April 2025*

*We appreciate the comments raised by the two reviewers, and thank them for the time and energy they spent on our paper.* Their comments below are in black text*, and our replies to each comment are in blue italics. Any* text directly added to the manuscript *will be shown in green italics in our reply below.*

**Reviewer #1:**

This manuscript describes how to propagating uncertainties and determine the information content of quantities derived from multiple instruments, and it illustrates this technique with an example. It is well written and will be a valuable addition to this topic. I recommend that it be published after minor revisions. I have listed comments/suggestions below that I hope the authors consider to make the manuscript a bit more accessible to those not proficient in these topics.

*Thank you for taking the time to read our paper and provide these comments!*

As a general comment coming from a reader who is not as familiar in this way of thinking as the authors, terms such information content and DFS are not in my daily vocabulary, and some of the results are not intuitive. I would urge the authors to assist us in making these concepts, which I believe is important ones, more accessible, and to provide more physical explanations where possible. For instance, the definition of information content is not given until line 55, and DFS not until line 110. Only on line 194 did I see it clearly stated that the variability in these quantities is due to instrument differences (also stated on line 324). For instance, is it possible to show or discuss the contributions of the various measurements to DFS? These comments should not be construed as criticisms of the manuscript, which is well written and reads nicely, but as an appeal to make challenging and non-intuitive concepts easier to comprehend.

*Information content in retrievals can be a challenging concept to grasp. We point out that the first paragraph of the abstract does indicate what perfect information content would be in an observation. However, as we know many people struggle to really understand the concept of information content, the first 50 lines of the paper are set up to bring the reader into the idea. In the body of the paper, the first time we introduce the phrase "information content" is line 53, and then we immediately define it in lines 54-55.*

*Similarly, the first time we define degrees of freedom for signal (DFS) is at line 110; it was not important to understanding the material that appeared earlier in the paper. We did add a new sentence at line 116, after we have defined DFS, that states:* "In other words, the DFS quantifies

the information content in the retrieval for each variable that is being retrieved in the vector *x*."

*We are not sure how to reorganize the paper to help the reader more easily understand either information content or DFS beyond*

Specific comments:

Line 130: This would be a good place to start a new section.

*We agree – and created a new subsection 2.2 Case study: 13 June 2019*

Line 145: It is not obvious from the figures that the retrieved winds are larger above 1 km.

*We believe that the reviewer mis-read the line: we state that "the uncertainties in the retrieved winds from the E37 DL…two DLs, especially above 1 km."  We believe that this is easily seen in Fig 3 c1,d1, which is what we are pointing out here. (Note we will slightly adjust this explanation, after addressing another comment from this reviewer down below).*

Line 177: Addition of another equation or a bit more explanation of how Eq. 5 becomes Eq. 6 would help the reader who is not as familiar with this topic as the authors are.

*We have added another sentence to help address this after Eq 6: "Note the translation of Eq 5 to Eq 6 uses the fact that $K_x = \partial F / \partial x$ and that the covariance of x can be written both as $\sigma_x^2$ or as the matrix $S_x$."*

Line 179: The phrase "where the superscript T in this content represents matrix transpose" should be place near Eq. 1 where it first appears.

*Good point.  We added that phrase to the sentence right after equation 1.  We also left this phrase after Eq 6 also (just to be clear).*

Lines 180-195: These would be better places before line 150.

*Excellent suggestion, and we agree.  We moved that text that describes the instrument-level uncertainties to (the new) section 2.2, just after Figure 3. This aids the flow of the information content derivation from the 6 instruments after Equation 6.*

Line 194: The explanation "due to the differences in the instrument uncertainties at the different locations" should appear earlier.

*We agree.  This was done when we addressed the suggestion immediately above.*

Line 200: The statement "the cold air advection … has a lot of uncertainty" is vague; is it the magnitude of the cold air advection, the timing, the direction, or what? Perhaps (likely?) that is my lack of understanding, but "cold air advection" sounds like a process, not a quantity (such as

temperature advection or water vapor advection); thus, it is unclear what the uncertainty would refer to.

*Ah, that is a good point. We are referring to the magnitude of the cold air advection here. We have clarified that point in the text.*

Line 203: Show the location of the other two sites in Fig. 4.

*On the spatial scale of this map, which we chose so that the larger synoptic conditions could be examined, if we used yellow dots of the same size then the three sites would be virtually on top of each other. This brings us to a comment raised by the other reviewer, who asked that we reconfigure Figure 1 to be more useful with distance information.*

Line 210: The statement that the information content is approximately 5 seems impossible from the statement on line 112 that the information content is between 0 and 1, and those shown in Fig. 5 are less than unity. I may be (likely am?) confusing different quantities, but that merely demonstrates that a typical reader may be confused here, and that a bit more explanation would be useful.

*We agree; our language was not clear. We have modified that sentence to say: "the sum of the total information content from the surface to 3 km (i.e., $\sum_{z=0 \text{ km}}^{z=3 \text{ km}} \text{DFS}(z)$) is approximately 5 for T(z) and between 3-to-5 for q(z)."*

Line 234: Perhaps move the title of the section to line 245, as that seems to be where the example actually starts.

*The reviewer is talking about the start of Section 4. The second sentence is that section is discussing the DFS of the temperature and water vapor advection, and pointing it out in Figure 5, which is part of this example. So we think that the section is starting in the correct place.*

Line 244: The statement that the information content on one quantity can be near unity even though the information content of an instrument can be low is crucial and should be more strongly emphasized.

*We agree. We have added the statement: "this is because the advection is essentially an evaluation of spatial gradients, which the AERI is able to determine even with its limited information content in the vertical."*

Line 244: "in information content" should be "an information content"

*Yes. Good catch*

Line 244: Panels b1, b2, and b3 of Fig. 5 show DFS of water vapor exceeding 0.05 at heights greater than 50 m.

*We updated that sentence to say: "AERI's information content is very limited with DFS < 0.05 at any height above 50 m for T and DFS < 0.3 for q"*

Line 245: The end of this sentence is a great location to remind the reader that this is due to the instrument.

*Good suggestion: we added "due to the larger instrument noise level in the E37 DL".*

Line 259: I had a "why?" after the statement that doubling the covariance matrices by a factor of 2 had little effect on the DFS, and found myself desiring a more physical explanation of this result.

*It is a curious result. We have added these sentences to offer some thoughts: "Presumably, this is because advection is a spatial calculation, and that the uncertainties at the vertices has relatively little impact on the derived advection. However, this result likely would depend on the size of the polygon used for the calculation; Wagner et al. (2022) demonstrated that the current locations of the ARM site is close to optimal in minimizing both the random and sampling error in the calculation."*

Figure 5: In the caption, perhaps label that columns 1, 2, and 3 refer to E37, C1, and E39 so they don't have to look back to Figure 2 to find this information. The rest of the information on this panel (e.g., T, WV, U, V) are labeled, but the locations aren't.

*Done.*

Line 260: "diagonal" or "diagonal elements"?

*We have adopted your suggestion of diagonal elements.*

Line 298: A bit more discussion would be helpful here. What happens at those altitudes, where the standard deviation is greater than the mean?

*We have added the phrase "implying that there is marked variability in the DFS above this height from case-to-case" to the discussion in that paragraph to discuss the implications.*

Line 324: This statement should appear much earlier in the manuscript.

*We agree, and added a new sentence directly after Figure 3 that states: "Note that differences in the noise characteristics among the AERIs will result in differences in the retrieval uncertainties; this is also true for the Doppler lidar systems."*

Line 331: "strong function height" should be "strong function of height"

*Done*

Line 336: "derived from remote sensors" or "retrieved from remote sensors" or "obtained from remote sensors"

*We restructured the sentence to make it clearer: "This work demonstrates how to derive the information content of an observation that is derived from multiple remote sensing datasets."*

---

## Author Response (AR2)

*11 May 2025*

*We thank the reviewers for their positive comments and their continued effort to help us present this work in as clearly as possible.  Our replies to the concerns raised are in blue italics below.*

Dear authors,

the paper is nearly ready for final publication. Some issues still have to be resolved before acceptance of the paper. The explanation of the DFS given in the author's reply is confusing and needs to be improved.

The DFS is one of three possible definitions of the information content, according to Rodgers. In your reply, you describe the Shannon's information content, which varies between 0 and 1. Value of 0 at a given altitude means that your measurement does not provide any information for this altitude (retrieval usually returns the a-priori value), while 1 means full information content from the measurement.

The DFS, another definition of information content, specifies how many independent pieces of information you have in the profile. When sampling the altitude range of 6 km in steps of 1 km, DFS of three means that on average the vertical resolution is about 2 km ( 2 km x DFS = 6 km). The width of the averaging kernels provides an estimate of the vertical resolution at each altitude,

It seems that you refer to different definitions of the information content (DFS, Shannon's information content) in your paper, but numerically use only DFS. You should clearly indicate in the paper which information content definition you refer to whenever you mention "information content".

*We apologize for the confusion in our manuscript.  The information content used in our analysis is the degrees of freedom for signal defined as the diagonal elements of the averaging kernel matrix A.  A different approach would be to use the Shannon Information Content (H; Shannon 1948), which is a measure of how the entropy of the system changes when the observations are included.  H is defined as*

$$H = -\frac{1}{2} ln\left(\frac{S_x}{S_a}\right)$$

*This measure of information content is somewhat similar to our Equation 7, which is*

$$A = I - \frac{S_x}{S_a}$$

*We have updated the manuscript to indicate that there are other measures of information content, and that we are using the DFS approach (which we then explicitly indicate will be discussed in Section 3).*

*The editor also made a couple of comments that we have addressed. We have updated our references to use the proper formatting, and have removed the reference to the paper that was submitted.*

---

## Author Response (AR3)

*13 May 2025*

*We have changed the comment on Figure 1 to state "© Google Earth" as suggested. There are no other changes.*

*Note: Figures 9 and 10 are two-part figures. We have provided individual EPS files for the left- and right-hand panels in the attached ZIP file (along with the other images).*

*Thank you.*